# Close Related Drug-Resistance Beijing Isolates of *Mycobacterium tuberculosis* Reveal a Different Transcriptomic Signature in a Murine Disease Progression Model

**DOI:** 10.3390/ijms23095157

**Published:** 2022-05-05

**Authors:** María Irene Cerezo-Cortés, Juan Germán Rodríguez-Castillo, Dulce Adriana Mata-Espinosa, Estela Isabel Bini, Jorge Barrios-Payan, Zyanya Lucia Zatarain-Barrón, Juan Manuel Anzola, Fernanda Cornejo-Granados, Adrian Ochoa-Leyva, Patricia Del Portillo, Martha Isabel Murcia, Rogelio Hernández-Pando

**Affiliations:** 1Laboratorio de Micobacterias, Departamento de Microbiología, Facultad de Medicina, Universidad Nacional de Colombia, Bogotá 111321, Colombia; micerezoc@unal.edu.co (M.I.C.-C.); jrodriguez@unal.edu.co (J.G.R.-C.); 2Sección de Patología Experimental, Departamento de Patología, Instituto Nacional de Ciencias Médicas y Nutrición Salvador Zubirán, Ciudad de México 14080, Mexico; dulmat@comunidad.unam.mx (D.A.M.-E.); estalabini@yahoo.com.ar (E.I.B.); jorge.barriosp@incmnsz.mx (J.B.-P.); zyanyal@hotmail.com (Z.L.Z.-B.); 3Grupo de Biotecnología Molecular, Grupo de Bioinformática y Biología Computacional, Corporación CorpoGen, Bogotá 110311, Colombia; juan.anzola@corpogen.org (J.M.A.); pdelportillo@corpogen.org (P.D.P.); 4Universidad Central, Facultad de Ingeniería y Ciencias Básicas Bogotá, Bogotá 100270, Colombia; 5Departamento de Microbiología Molecular, Instituto de Biotecnología, Universidad Nacional Autónoma de México, Cuernavaca 62210, Mexico; fernanda.cornejo@ibt.unam.mx (F.C.-G.); adrian.ochoa@ibt.unam.mx (A.O.-L.)

**Keywords:** *Mycobacterium tuberculosis*, lineage 2/Beijing, in vivo transcriptomics, murine model, RNAseq, virulence

## Abstract

*Mycobacterium tuberculosis* (MTB) lineage 2/Beijing is associated with high virulence and drug resistance worldwide. In Colombia, the Beijing genotype has circulated since 1997, predominantly on the pacific coast, with the Beijing-Like SIT-190 being more prevalent. This genotype conforms to a drug-resistant cluster and shows a fatal outcome in patients. To better understand virulence determinants, we performed a transcriptomic analysis with a Beijing-Like SIT-190 isolate (BL-323), and Beijing-Classic SIT-1 isolate (BC-391) in progressive tuberculosis (TB) murine model. Bacterial RNA was extracted from mice lungs on days 3, 14, 28, and 60. On average, 0.6% of the total reads mapped against MTB genomes and of those, 90% against coding genes. The strains were independently associated as determined by hierarchical cluster and multidimensional scaling analysis. Gene ontology showed that in strain BL-323 enriched functions were related to host immune response and hypoxia, while proteolysis and protein folding were enriched in the BC-391 strain. Altogether, our results suggested a differential bacterial transcriptional program when evaluating these two closely related strains. The data presented here could potentially impact the control of this emerging, highly virulent, and drug-resistant genotype.

## 1. Introduction

There is increasing evidence that, in addition to environmental factors and human genetics, strain variation in the *Mycobacterium tuberculosis complex* (MTBC) plays a role in the outcome of TB infection and disease [1,2,3]. According to the presence or absence of the tuberculosis-specific deletion region and whole-genome sequence analysis [4,5], the MTBC have been grouped into nine lineages classified as “ancient” (lineages 1, 5, 6, 7, 8, and 9) or “modern” (2, 3, and 4) [6,7]. The impact of these MTB lineages on the outcome of mycobacterial infections, although scarce, has been reported, i.e., lineage 3 strains were associated with reduced risk of transmission chains in Europe and the Americas. In contrast, lineage 2/Beijing strains were associated with treatment failure [8]. Furthermore, contacts exposed to lineage 2, in which the Beijing family is the most representative, were most likely to progress to disease [9]. Also, this lineage has been associated with higher virulence [10].

The lineage 2/Beijing strains have been classified as “ancient” or “modern” according to different genetic polymorphisms [11]. The deletion of the region of differentiation (RD) RD105 is a hallmark of the lineage 2/Beijing, RD207 and RD181 are markers of “ancient” Beijing while the modern Beijing are characterized by the IS6110 insertion sequence in the Noise Transfer Function (NTF) gene, and RD142 and RD150 [12]. According to the Spoligotyping pattern, the Classical Beijing strains are characterized by the absence of the spacers 1 to 34 and strains with the absence of additional spacers between 35 and 43 are classified as atypical or Beijing like [13]. Although, a low prevalence of the Lineage 2/Beijing strains have been reported in South America [14,15] in Cuba, Perú, and Colombia, a high prevalence of multidrug-resistant (MDR) isolates has been reported [16,17]. During the last decade, we have identified an extensively drug resistance (XDR)-MTB Beijing genotype emerging in Colombia, especially in the Southwestern region of our country [15,17,18]. Different isolates from this genotype have shown that they are XDR and highly aggressive due to a high mortality rate [16,17]. Spoligotyping analysis classifies this genotype as Beijing-like (BL) with a pattern of SIT 190 [17]. In previous work, we performed whole genomic sequencing of six representatives of these BL isolates and did comparative genomic with the Beijing Classic (BC) strains SIT 1 that also circulated on Colombia’s pacific coast [19]. Regions of deletion RD105, RD149, RD207, and RD181 present in BC strains are also present in the BL strains from Colombia. They all belonged to the modern Beijing strains, constitute a cluster and share specific nucleotide variants (SNPs and deletions) associated with virulence and XDR [19].

One of the biggest challenges in the immunopathology of TB is the link between the genetic variability of the pathogen with the clinical outcome of TB. The murine model of tuberculosis infection is beneficial for understanding the pathogenesis of the bacillus as well as the immune response of the host. The expression of genes related to the host’s immune response has shown key immune system factors for disease control such as interleukins, IL-12, IFN γ, Tumor Necrosis Factor (TNF-α), and IL-1, among others [20]. However, the study of pathogen’s gene expression in response to the host environment in murine models is more complex to obtain since the bacillus-host cell messenger RNA ratio can be as high as two orders of magnitude. Nevertheless, the increased sensitivity of transcriptomic analysis has allowed conducting in vivo transcriptomic RNAseq protocols permitting in the same experiment to study both the host and the pathogen responses [21].

To understand the differences in the pathogenesis of the BL and BC closely related strains, we conducted a transcriptomic assay with two representative strains of these genotypes, the BL-323 and the BC-391. Both strains have 14 IS6110 insertions located at the same position in the chromosome, but the strain BL-391 has an extra copy located between EsxR and EsxS. In addition, the strain BL-323 differs in 9 SNPs, including five missenses (*mmsA*, *ubiA*, *whiB6*, and two unknown genes); three synonymous (Rv0197, Rv2308, and Rv2940), and a deletion of 30 amino acid in the PPE8 protein [19]. In a previously reported work, the lung transcriptome of mice infected with either Beijing strains was performed [22]. In the present work, we report the analysis of the bacterial transcriptome in the same lungs from this murine model.

Applying the recently developed strategy to enrich the mRNA of the bacillus [23], we carried out a transcriptional study of the BL-323 and the BC-391 directly in a murine disease progression model. Our results show a different transcriptional program between these two closely related Beijing strains, indicating that minor genetic variations among MTB strains surprisingly induce a different immunological response, but in the case of these strains, without accurate control of the pathogen. Our results gave us insights into the pathogenesis of this deadly pathogen. This work could lead to the discovery of possible therapeutic targets for the treatment and control of TB.

## 2. Results

### 2.1. Infection of Mice with MTB Beijing Strains

To have a better understanding of the virulence determinants associated with the BL genotype circulating in Colombia and to gain insights into the intricate interactions between the host defense systems and the specific systems regulating mycobacterial gene expression, we performed in vivo transcriptomic assays in BALB/c mice infected with the XDR BC-391 and the XDR BL-323 strains. In the same experiment, we conducted two types of analysis: the transcriptomic response of the lung mice infected with either strain that was already published [22], and in the present work the transcriptome of both MTB strains was determined, following a new method to enrich bacterial isolation from the infected mouse lung obtaining RNA in proper and quality amount to determine the bacterial transcriptomic profile [23]. Thus, for the first time, the MTB Beijing strains transcriptome in vivo is determined.

As previously shown, the BL-323 strain has a more virulent behavior in the murine model than the BC-391 strain. The virulent phenotype was characterized by a high bacillary load and early death of animals [22]. Based on the lung histopathology, samples for RNAseq experiments were removed from the lung of infected mice at D3, D14, and D28 PI for the BL-323 strains and at D3, D14, D28, and D60 PI for the BL-391.

### 2.2. Global Transcriptomic Analysis

On average, a total of 11 million reads were obtained in the transcriptomic studies, ~7.6% of these did not map against the mouse genome and were analyzed against the BL-323 (NCBI: CP010873) and BC-391 (NCBI: CP017596) reported genomes (Table 1 and Appendix A). The complete sequence project was deposited in the GEO database, Accession Number: GSE 198877 https://www.ncbi.nlm.nih.gov/geo/query/acc.cgi?acc=GSE198877 (accessed on 24 March 2022).

Of the total reads, 0.6% mapped to MTB genomes. Of these, 90% mapped against coding sequences (CDS), indicating that the strategy applied to the mRNA hybridization was successful in the remotion of bacterial rRNA as can be observed in Table 1 and in the enrichment of bacterial mRNA. As expected, for a closed related strain, a very similar percentage of reads mapped against the open reading frames (ORF) in the different functional categories of MTB (Figure 1). This is observed in both strains and at all time points of the infection. Approximately 1200 genes of MTB had at least two reads, which demonstrated that our data covered the 33% of the genes from the total genome [24] (Table 1), the remaining genes were not detected in our transcriptomic assay. The 7% of reads that did not map either to the mouse or MTB genomes were analyzed by METAPHLAN 3 [25]. None of our sequences matched this database, and therefore we concluded that these sequences are artifacts generated during the library construction.

### 2.3. Differential Expression Analysis

We carried out two types of analysis of differentially expressed genes: (i) intra-strain analysis to identify genes involved in the response of each strain to the stress imposed by the host’s immune system during the infection process and (ii) inter-strain analysis to identify those genes that could be associated with the virulence phenotype observed between the strains. All days PI in both strains were analyzed.

### 2.4. Intra-Strain Analysis

The list of genes differentially expressed between days PI for both strains are presented in Appendix A. In addition, a landscape of expression was performed by heatmap analysis and Gene Ontology Analysis (GOA) along the infection process in both strains.

### 2.5. Strain BL-323

The strain BL-323 shows few genes overexpressed at D3 and D14, this scenario changes at D28 with an increase in the number of genes overexpressed, as can be seen in the heatmap of Figure 2A.

Strain BL-323 GOA of the differentially expressed genes on D14 vs. D3 shows mostly genes related to response to host immune system and hypoxia (Figure 2B). In BL-323, 15 genes are overexpressed on D14 while no one is under-expressed on the same day PI. Overexpressed genes are involved in metabolism (*pckA*), electron transport (NADH dehydrogenase, *ndh*), ferredoxin, toxin–antitoxin system (*vapC16*), and response to hypoxia (*hspX* and *Hrp1*). The response regulator *trcR* (Rv1033c), belonging to the two-component system TrcR/TrcS [26] is also overexpressed at D14. This response regulator represses the expression of the gene *Rv1057*, which affects MTB’s intracellular growth and reduces the expression of ESAT-6 [27]. Macrophages infected with a MTB *Rv1057* deleted mutant reduce the secretion levels of cytokines IL-1β, IL-10, TNF-α, and IF-gamma, which have been shown to induce a pro-inflammatory response [26]. GOA at D28 vs. D3 maintains similar functional features as D14, which means response to immune system and hypoxia (Figure 2B). Analysis of DEG between D28 vs. D3 shows that 8 of the 15 genes overexpressed on D14 remain highly expressed on D28, including *hspX*, the *Hrp1*, and *pckA*, but not the response regulator *trcR*, which is down-regulated (Appendix A). We also identified 33 new overexpressed genes, including the transcription factor *sigB*, the secretion protein EspA and the ABC transporter *Rv3197* responsible for the resistance to erythromycin [28] (Appendix A). Although most of the small non-coding RNAs were removed in the library construction (fragments small of 200 bp), the presence of sRNA MTS2823 at D28 PI is remarkable. The MTS2823 is involved in the regulation of transcription at stationary phase [29], during growth in lipid environment [30], and in vivo infection [31]. Additionally, we detect the expression of the sRNA mcr7 responsible for downregulating the TAT secretion system throughout the *phoP* transcription factor [32]. Also, mcr7 completely overlaps with the gene *aprA* (Acid and phagosome regulated protein A) [33] (Appendix A). On D28 vs. D14, GOA shows mostly genes related to unfolded protein binding and response to hypoxia, although most of the genes that contribute to these functions were more expressed on D14. Only three genes were overexpressed on D28 compared to D3 (Figure 2B). One gene was the *ftsH*, which codifies for an essential membrane-bound protease that degrades integral membrane proteins and cytoplasmic proteins. Increased expression of this gene delays growth and reduces the viability of MTB in vitro and ex vivo experiments [34]. Venn diagram of the DEG along the days PI for strain BL-323 was shown in Appendix A.

### 2.6. Strain BC-391

Unlike the strain BL-323, the intra-strain analysis of BC-391 shows a more active transcription profile at D3 PI (Figure 3A and Appendix A). At D14 vs. D3 PI, GOA analysis shows mostly genes related to protein binding, proteolysis, and oxidoreductase activity (Figure 3B). There are only two genes overexpressed at D14; they are the *hemA*, and the *ahpC* essential for the pathogen’s survival inside macrophages [35].

Two sRNAs were overexpressed at D14. The most expressed transcript was the sRNA mcr7 that, as mentioned above, controls the TAT secretion system [32]. It is followed by the sRNA Asdes, involved in regulating the mycolic acid biosynthesis [36]. Although interestingly, the transcription observed on D3 decreases on D14, we do not have an explanation for this behavior; however, we cannot ignore the expression of the sRNAs on D14.

At D28 vs. D3, there was 88 DEG (Appendix A), with only four being overexpressed, among them, the *uspA* (Rv2005c) gene whose function is unknown but belongs to the DOS regulon, and recently this protein showed induction of dendritic cell maturation and Th1 stimulus in macrophages [37].

Finally, the comparison between D28 vs. D14 shows 128 DE genes, with 56 overexpressed in D28 (Appendix A). GOA analysis display genes predominantly related to the phosphopantetheine binding activity included in several acyl carrier proteins: fatty acid synthases; polyketide synthases; peptidyl carrier proteins, as well as aryl carrier proteins of non-ribosomal peptide synthetases [38]. The second most abundant function is the long-chain fatty acid metabolic process. The most expressed genes are *Rv0791c*, belonging to LLM class F420-dependent oxidoreductase; the second is *Rv0823c*, a probable tRNA dihydrouridine synthase *dusB*, involved in tRNA modification during stress conditions [39]. Additionally, the alternative sigma factor *sigG* was expressed, probably involved in detoxifying methylglyoxal [40]. In this sense, another overexpressed gene related to detoxification is the *Rv1280c*, an oligopeptide ABC transporter *oppA* involved in the uptake of glutathione [41]. Finally, at D60 vs. D28, only one gene is over-expressed, the *Rv1692* involved in converting glycerol-3-phosphate (G3P) to glycerol. A Venn diagram of the DEG along the infection is shown in Appendix A.

### 2.7. Inter-Strain Analysis

The intra-strain analysis describes above indicates a different transcriptomic program during the infection strain-dependent. This finding was reinforced by the results obtained with the inter-strain analysis. First, we conducted a global analysis of the data to observe how the samples group between the strains. Then, we performed a Multidimensional Scaling Plots (MDS) analysis, which determines the most significant sources of variation in the data. As can be seen in Figure 4A, the more significant source of variation (44%) is the strain and not the day PI (17%) (Figure 4B). This result is surprising given the high genetic homology between BL-323 and BC-391 and supports the view of the distinctive transcriptomic landscape between these two closely related genetic strains.

We performed a heatmap to visualize the differentially expressed genes and the transcription profile between the strains. As shown in Figure 5, we observed a different transcription pattern between the analyzed strains.

Differentially expressed genes between strains at the time points are shown in Appendix A. At D3, only 15 genes were DE strains; the number of DEG increased with the days PI. At D14, 81 genes were DE, and at D28, 161 genes were DE. Some genes were shared between days but had different expression patterns (Appendix A). These results indicate that during infection, their differences are increasing. At D3 in the strain BC-391, we found among the overexpressed genes, the PPE 38 (*Rv2352c*), which has been demonstrated to inhibit the MHC-1 presentation for macrophages [42], the *Rv0004*, involved in the replication of DNA [43], the Ppr1 (*Rv3333c*) which control the sigma factor SigM by hydrolysis of its Anti-SigM factor [44], the *pknB* (*Rv0014c*) which is involved in the peptidoglycan synthesis and maintenance of cell growth [45], *Rv1500* responsible of synthesis of phosphatidylinositol mannosidase and lipoarabinomannan [46], and the secretion protein EccB belonging to ESX-5 secretion system [47]. This transcriptomic profile indicated an active growth and probably mechanisms of immune evasion. There are only five overexpressed genes in the BL-323 strain, two of them are hypothetical proteins, and none have known function (Figure 5). On D14 PI, the transcriptomic profile changes dramatically. Only *pknB* and *Rv1263,* a putative amidase, are overexpressed in strain BC-391.

Conversely, 79 genes were overexpressed in strain BL-323. Genes related to the ESX-1 secretion system, *Rv1388* (*mIHF*), regulated proteins involved in the type VII secretion system, the small RNA *mcr7*, which regulates the TAT secretion system, the transcription factor *trcR*, were all overexpressed on this day. The expression of these genes could suggest an active growth and expression of virulence factors in the BL-323 strain. On D28 PI, 38 genes were overexpressed in strain BC-391, and 123 genes were overexpressed in strain BL-323. Among the most expressed genes in BC-391 is the gene *lpqM,* a lipoprotein required for survival in primary murine macrophages [48] and increased under starvation in vitro [48]. Another gene expressed in BC-391 is Ag85A which allows bacteria to evade the host immune response by preventing the formation of phagolysosomes [49]. On the other hand, in the strain BL-323, the most exciting genes are the *eccB* gene belonging to the ESX-5 secretion system, the sigma factor *sigE,* and the gene *lytR*, which is essential in innate immune evasion and virulence [50,51].

## 3. Discussion

One of the main aspects of TB control is understanding the genomic variability among different strains of MTB and its association with the outcome of TB disease. Therefore, the knowledge of MTB genes involved in virulence and their response against the host immune attack is of pivotal importance. In this study, we wanted to identify those genes expressed in two closed related strains of the MTB Beijing genotype at the site of infection in murine models using the recently RNAseq technology developed by our laboratory [23]. Both strains have shown to be highly virulent, with BL-323 more virulent because it produced a higher mortality rate in infected mice, more pulmonary bacillary burden, and tissue damage than those induced by BC-391 [17,22].

RNAseq technology has been used in multiple assays, especially in the analysis of different conditions of growth in vitro [52] or ex-vivo [53]. However, the assays of RNAseq in vivo, i.e., murine models or specimens of tuberculosis patients, are scarce [54]. The challenges in generating RNAseq datasets from in vivo material in which bacterial burden is low and variable and host cell heterogeneity is high, remain daunting. We used the strategy reported by Cornejo et al. to enrich the mRNA from BL-323 and BC-391 strains [23]. In their work, the strain H37Rv was used in the murine model finding that 1.7% of the total reads mapped to MTB; of these, 68% mapped against CDS and 32% against intergenic regions. In our study, 0.6% of the total reads mapped against MTB, but 90% of these reads mapped against CDS. This difference was probably due to the different strains used and the type of the study [23].

The MTB transcriptomics reported here are part of a broader experiment in which simultaneous samples were taken to make the transcriptomics of the mouse’s immune response. These results were reported in a previous article [22]. In this sense, one of the most exciting results is that the strains induced a transitory different immune response in syngeneic mice, depending on the time PI. The BL-323 strain seems to induce an anti-inflammatory response during the first 14 days of the TB infection (Figure 6). Two genes overexpressed at this day could be key players for this response, the *hrp1* and the *trcR* genes. The *hrp1* blocks two critical proteins involved in activating the transcription factor NFkB, which triggers the pro-inflammatory response in macrophages [55]. The *trcR* response regulator represses the expression of *Rv1057* suggesting that the secretion levels of cytokines like IL-1β, TNF-α, and IFN-γ involved in the pro-inflammatory phenotype are reduced.

On the other hand, the Esx1 secretion system is also affected by the repression of the *Rv1057*. This anti-inflammatory profile agrees with that observed in mice infected with this strain since the response to infection on D 14 PI is predominantly anti-inflammatory [22]. However, this scenario changed on D28 PI when we observed the downregulation of the *hrp1* and the *trcR* genes and overexpression of genes associated with the Esx1 secretion system, probably leading to a burst of antigens related to virulence (Figure 6). The repression of the *trcR* gene on D28, probably due to their autoregulation [56] could be associated with the increased expression of genes associated with secretion. These changes indicate the induction of the pro-inflammatory response mediated by the activation of the Esx1 secretion system. This scenery correlates with the immune response observed in mice infected with this strain on day 28PI, producing extensive pneumonia and the animal’s death by respiratory insufficiency [22].

In the case of the strain BC-391, a different scenery was observed on D14 PI. The sRNAs mcr7 and Asdes were the most expressed transcripts on this day. As mentioned before, mcr7 negatively regulates the TAT secretion system and, therefore, impacts the secretion of diverse proteins, Ag85A, and PPE38, two antigens that stimulate the immune system. On the other hand, ASdes is antisense of mRNA coding for proteins DesA1 and DesA2 involved in the mycolic acid biosynthesis. Interestingly the repression of genes controlling bacterial growth like *clpX* and *clpC1* proteases indicated an active growth on day 14 PI. Mice infected with this strain showed expression of genes related to pro-inflammatory response on D14 PI [22] but this response is not due to protein secretion mediated by the TAT system (Figure 7). On D28 PI, 56 genes are overexpressed compared with D14 PI. Some functions related to these genes (siderophore and mycobactin synthesis, ABC transporters, detoxification, aminopeptidases, DNA reparation, and long-chain fatty acid synthesis) indicate an adaptation of the bacilli to the stress imposed by the macrophage attack (Figure 7).

One of the most critical genes is the *oppA* which codifies for an ABC transporter protein involved in the glutathione uptake [41,57]. Dasgupta et al. (2010) demonstrated that the uptake of glutathione by the bacilli impairs the capacity of the infected macrophages to detoxify methylglyoxal (MG), a physiological product of various metabolic pathways. Increased MG production is observed in the granuloma and is associated with elevated inflammatory cytokine production and apoptosis [41]. Additionally, the induction of the transcription factor *sigG* confirms that the bacillus is undergoing a detoxification process on D28 [58]. At this moment, mice infected with this strain showed lesser pneumonia and production of pro-inflammatory cytokines compared with mice infected with BL-323. However, there was extensive pneumonia and necrosis at the late stage of infection, probably due in part to the toxicity of MG [41,57]. These results show that the two strains follow a different transcriptomic program and induce a transitory different immune response in the mice model.

From the inter-strain comparison point of view, the two strains have different transcriptomes programs along the infection (Figure 4), confirming the results obtained for individual strains (intra-strain analysis). At the beginning of infection, we can infer two aspects regarding the DE expression genes of the BC-391 strain, one related to manipulation of the immune system inhibiting the presentation by the MHC1 and at the same time allowing the secretion of PE/PPE proteins by the *eccB* gene. Expressed genes related to the synthesis of cell wall lipids and DNA replication indicates active growth. On the other hand, the BL-323 shows other genes related to the synthesis of PDIM and PE proteins indicating a different stimulus to the immune system. On D14 PI, both strains make drastic changes. The DEG by the BL-323 strain indicate the induction of virulence factors, secretion of PE/PPE proteins, and metabolic processes related to adaptation to intracellular stress imposed by macrophages. Despite this bacterial profile, the mice infected with this strain showed a predominant anti-inflammatory response. In the BC-391 strain, only two genes are DE, the *pknB* responsible for maintaining bacilli growth, and an amidase of unknown function.

On D28 PI, DEG in BC-391 suggested adaptation to intracellular stress, secretion of PE/PPE proteins, and lipid synthesis. We suggest that an essential mechanism of beginning tissue damage is the *oppA* expression on this day since it captures glyoxylate and stimulates the apoptosis mechanism, as mentioned before [44] (Figure 8). In the strain BL-323 on D28 PI, more DEG are obtained regarding the BC-391 strain. This group of genes is related to the Esx-1 secretion system, chaperones, and survival to intracellular stress. Remarkable is the presence of the extracytoplasmic factor sigma *sigE* whose expression is essential for virulence in macrophages and in blocking phagosome maturation, thus leading to more-efficient antigen presentation [59]. This important sigma factor was recently used in a development vaccine with promising results [59]. All of these coordinated orchestrated functions would be responsible for the greater virulence of the BL-323 strain concerning the BC-391 strain observed during infection in the mice model.

We cannot ignore the mutations reported in the BL-323 strains *mmsA*, *ubiA*, *whiB6*, and two unknown genes, which could also be responsible for a more virulent phenotype observed in this strain. Significantly, the missense SNP in *mmsA*, which interferes with IFN-1 response and the missense in the transcriptional regulator *whiB6*, which, as shown previously in another Beijing strain [60] could induce a more virulent phenotype.

Even though more effort should be made to refine the techniques used for pathogen/host transcriptomic assays or dual RNA-seq experiments, this paper shows the importance of this kind of assays in trying to understand how genomic variability among a clonal population like MTB could impair the host immune response. This knowledge will pave the future of new anti-tuberculosis drugs and diagnostics. We hope other dedicated approaches to time-course analysis will be forthcoming.

## 4. Materials and Methods

### 4.1. Ethics Statement

Animal experiments were performed following ethic procedures according to the ethical criteria of Universidad Nacional de Colombia Faculty Medicine ethics committee, Assessment Act 011-104-15 and the Mexican Law NOM 061-Z00-1999, approved by the Internal Committee for the Care and Use of Laboratory Animals (CICUAL) of the Instituto Nacional de Ciencias Médicas y Nutrición Salvador Zubirán (Protocol number PAT-1846-16/20).

### 4.2. Bacterial Strains

The present study used two MTB strains belonging to the East Asia lineage 2 Beijing family isolated from Colombian patients. The XDR BL-323 strain, SIT 190 [17] was isolated from a female fifteen-year-old patient from the Port of Buenaventura on the Colombian west pacific coast. The MDR BC-391 strain was kindly donated by the Mycobacteria Laboratory from the National Institute of Health in Colombia (INS), Bogotá, Colombia. Bacterial genotype was confirmed by Spoligotyping and MIRU-VNTR 24-loci [61]. Antibiotic susceptibility profile was carried out as previously described [17,19].

### 4.3. Bacterial Culture

Clinical isolates were cultured in Middlebrook 7H10 supplemented with 5% glycerol. For the infection of mice, an isolated colony was grown in 60 mL of Middlebrook 7H9 medium supplemented with 5% glycerol, 10% OADC, and 0.02% of Tyloxapol Premium (Cat No: 25301-02-4, Merck, Germany) without the addition of antibiotics. When bacteria reached the mid-logarithmic growth phase (0.013 for BL-323 and 0.0325 for BC-391 at OD 680), the bacteria were centrifuged at 3500 rpm, and aliquots of 1 mL were stored at −80 °C until use. For mice infection, each aliquot was evaluated for bacterial concentration by UFCs. For mice infection, 250,000 UFCs/mL in sterile saline solution were used.

### 4.4. Infection of Mice with MTB Beijing Isolates for Transcriptomic Assays

The progressive pulmonary TB model [62,63] was used for the transcriptomic assay of Beijing isolates. The bacterial aliquots were thawed and sonicated for 30 s at 20 kHz to disaggregate the bacterial clumps and held at 4 °C until inoculation. Male BALB/c mice, between 6 and 8 weeks old with ~22 gr weight were used. Mice were anaesthetized with 100% Sevofluorane in a gas chamber and infected with both Beijing isolates by trachea exposure using a total of 250,000 bacteria. After spontaneously recovering, six animals were housed in cages with microisolators in a vivarium with level III safety facilities. The kinetics of progression of infection were previously reported [62]. Lungs of 3 animals (considered biological replicates) were used for RNA extraction and purification to perform the RNAseq experiments of the Beijing isolates.

### 4.5. RNA Extraction

Lungs stored at −80 °C were removed and maintained in liquid nitrogen until processing. All of the procedures were handled at 4 °C to avoid RNA degradation. Both mouse lungs were pulverized with liquid nitrogen, and aliquots of lung powder were stored to −80 °C until used. To release the bacterial cells, mild-lysis buffer RLT Plus (Cat No: 1053393 Qiagen, Valencia, CA, USA) supplemented with 10% β-mercaptoethanol was added. After centrifugation at 14,000 rpm at 4 °C for 5 min, the bacterial enriched pellet was used for RNA extraction using the Quick-RNA Miniprep Kit (Cat No: R1054, Zymo Research, Irvine, CA, USA), following the manufacturer’s instructions. Quantity and quality of isolated RNA were verified by 260/280 ratio using the EpochTM Microplate Spectrophotometer, Agilent, CA, USA, and the integrity of ribosomal subunits was evaluated in 2% agarose gel. Verifying ribosomal subunits integrity (RIN) was performed with the microchip bioanalyzer Agilent, following the manufacturer’s instructions (Agilent RNA 6000 Nanochip Cat No: 5067-1511, Santa Clara, CA, USA). Samples with RIN higher than 7 were used to construct RNA libraries. The quantification of RNA for libraries construction was carried out with Qubit RNA HS (Cat No: Q32852, ThermoFisher, Waltham, MA, USA).

### 4.6. Eukaryotic Ribosomal RNA Depletion and Libraries Construction

Depletion of eukaryotic ribosomal subunits was carried out with the Ribo-Zero Kit (llumina^®^ Ribo-Zero Plus rRNA Depletion Kit Cat No: 20037135, San Diego, CA, USA) following the manufacturer’s instructions. Depleted RNA (500 ng of each sample) was used to construct libraries using the TruSeq^®^ Stranded mRNA llumina^®^ kit (Cat No: 20020594, San Diego, CA, USA). The fragmentation time was adjusted to 14 min to obtain ~200–400 bp inserts, and the PCR enrichment was set to 15 cycles. To ensure good quality and quantity for the libraries, the concentration was verified by Qubit™ dsDNA HS (Cat No: Q32852, ThermoFisher, Waltham, MA, USA), and the size of the libraries was verified using Agilent 2100 bioanalyzer System High Sensitivity DNA Kit (Cat No: 5067-4626, Agilent, Santa Clara, CA, USA). The libraries with the correct insert size were subjected to bacterial ribosomal subunits depletion with an in-house subtractive hybridization [23].

### 4.7. Bioinformatic Analysis

Quality control was performed using the tool MULTIQC (https://multiqc.info accessed on 1 June 2021) [64] were trimmed with TRIMMOMATIC (http://www.usadellab.org/cms/?page=trimmomatic accessed on 1 June 2021) [65], using 25 as quality filters and a length of 50 nucleotides. The mapping was done against the genome sequences of the same strains previously reported in the NCBI with the access numbers CP017596 for BC-391 (https://www.ncbi.nlm.nih.gov/search/all/?term=CP017596 accessed on 15 June 2021) and CP010873 for BL-323 (https://www.ncbi.nlm.nih.gov/nuccore/CP010873.1 accessed on 15 June 2021) using SMALT tool of Sanger Institute (https://www.sanger.ac.uk/science/tools/smalt-0 accessed on 1 June 2021) (SMALT is Copyright (C) 2010–2015 Genome Research Ltd.). A first mapping process was done against a mouse reference genome (Mus musculus assembly GRC m38.p6), saving the information in two separated files (one with reads mapped against mouse sequence and another file with reads that did not map against mouse); reads mapping to mouse were removed, the rest of them were subsequently mapped against MTB H37Rv and to the available BL-323 and BC-391 genomes. The raw data are presented in Appendix A.

An analysis with the tool Feature Counts of SUBREAD package (http://subread.sourceforge.net accessed on 15 June 2021) was performed to obtain the differentially expressed genes. The differentially expressed genes were obtained according to the number of reads mapped to each gene or non-coding region. Two or more reads were considered transcriptionally active in any DNA region and included in the transcriptomic analysis. Differentially expressed genes were determined using the tool edgeR (http://bioconductor.org accessed on 15 June 2021) [66]. Each strain was analyzed along the infection process at different day points (D). For example, D3, D14, and D28 post-infection (PI) were analyzed for the BL-323 strain, while D3, D14, D28, and D60 PI were analyzed for the BC-391 strain. For each comparison, an FDR ≤ 0.05 was used as a cut-off to determine differentially expressed genes, the magnitude of the difference in expression was determined, and graphs were constructed to represent a change in magnitude for each gene.

### 4.8. Gene Ontology and Functional Enrichment Analysis

To associate genes with functions, we performed a Gene Ontology Analysis (GOA) using The Gene Ontology portal (http://geneontology.org accessed on 6 October 2021) [67]. Based on the ontologies, it is possible to make a functional mapping of genes of an organism. Finally, a functional enrichment analysis was performed based on the ontology results using the GeneMerge software (http://www.genemerge.net accessed on 10 October 2021) [68] to obtain the enriched molecular functions of the genes shown by the ontology analysis (*p*-value ≤ 0.05). The list of genes DE between comparisons is shown in Appendix A, with positive or negative values, where positive values indicate genes overexpressed on that day compared with the day of reference.

### 4.9. MDS Analysis

Multidimensional Scaling was used to visualize the most significant sources of data variation. Gene expression levels were transformed to log2-fold changes and visualized with the plotMDS function from the edgeR package.

### 4.10. Heatmap

Only differentially expressed genes were used to construct the heatmap. First, gene expression was normalized with the calcNormFactors function from the edgeR package. Next, gene distances were used to form clusters with the hclust function and the Pearson correlation coefficient. Code for this analysis can be found in the Appendix A.

### 4.11. Methaplhan (Metagenomic Phylogenetic Analysis)

In order to confirm if the non-mapped reads to the M. tuberculosis genome could come from the natural mouse microbiota, we decided to use METAPLHAN to see if those read matched to any other microorganism. This bioinformatics tool allows profiling the composition of microbial communities at the metagenomics level [25].

## Figures and Tables

**Figure 1 ijms-23-05157-f001:**
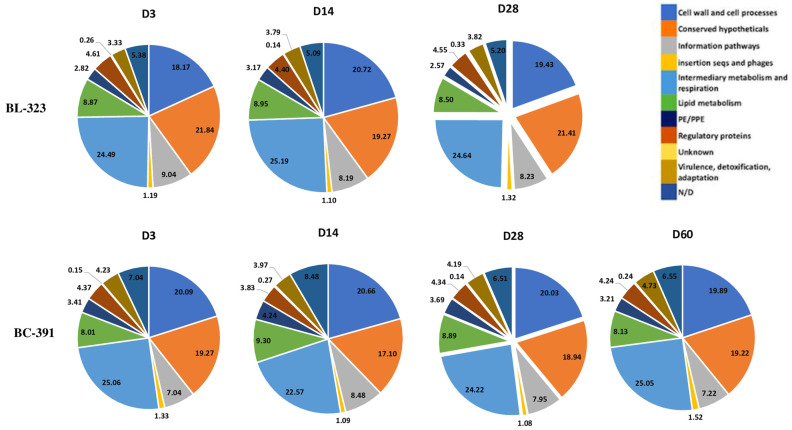
Percentages of mapped reads against the functional categories of MTB (https://mycobrowser.epfl.ch/ accessed 4 November 2021). The time point of the analysis is showed for both strains. D: day of infection.

**Figure 2 ijms-23-05157-f002:**
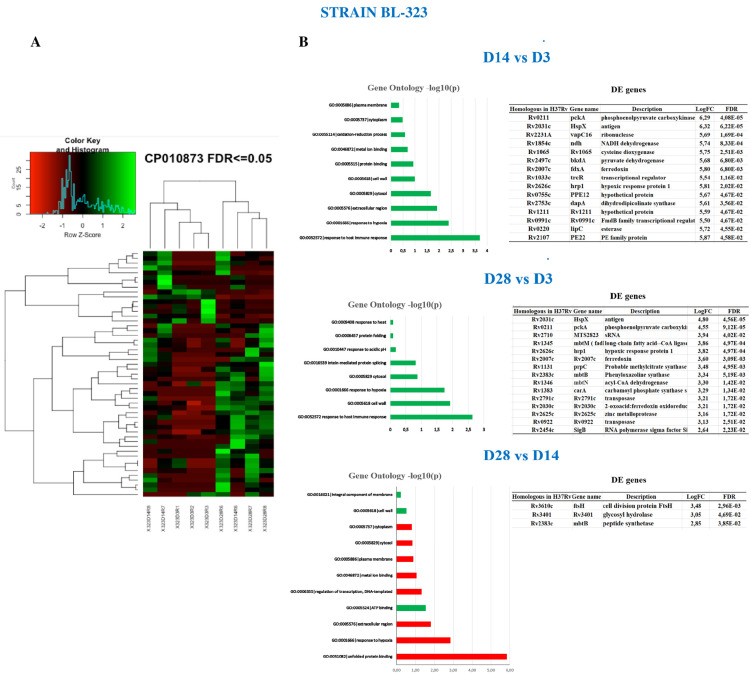
MTB strain BL-323 gene expression differences between days PI. (**A**) Heatmap showing the differentially expressed genes. Overexpressed genes are colored in green whilst under expressed ones are colored in red. Biological replicates were included in the analysis. (**B**) GOA at the different time points of infection (*p*-value ≤ 0.05). Green bars are metabolic pathways whose genes are overexpressed on the day of the assay, whilst red bars are metabolic pathways whose genes are more expressed on the day of reference. Only overexpressed genes are displayed in the tables.

**Figure 3 ijms-23-05157-f003:**
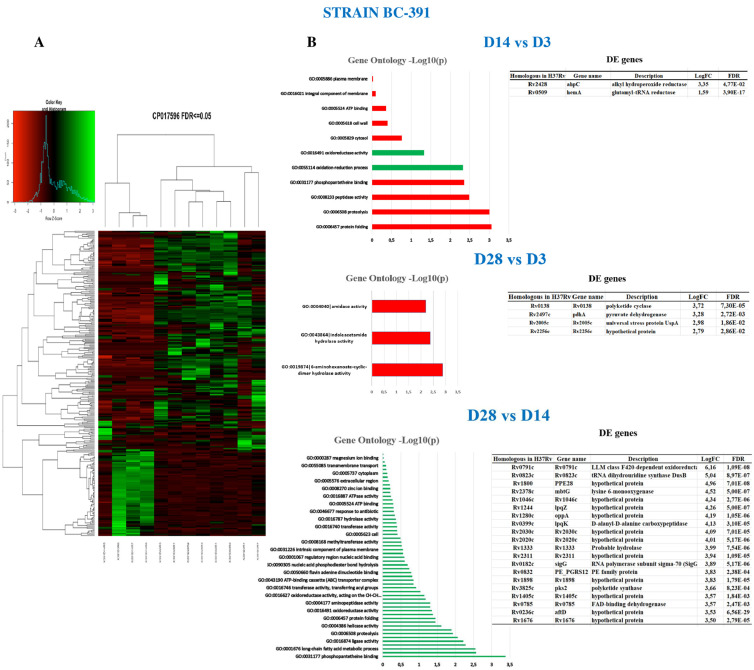
MTB strain BC-391 gene expression differences between days PI. (**A**) Heatmap showing the differentially expressed genes. Overexpressed genes are colored in green whilst under-expressed ones are colored in red. Three biological replicates were included in the analysis. (**B**) GOA at the different time points of the analysis (*p*-value ≤ 0.05). Green bars are the enriched metabolic pathways, while red bars are underrepresented. The overexpressed genes are displayed in the tables. D60 vs. D28 is not presented because only one gene was DE.

**Figure 4 ijms-23-05157-f004:**
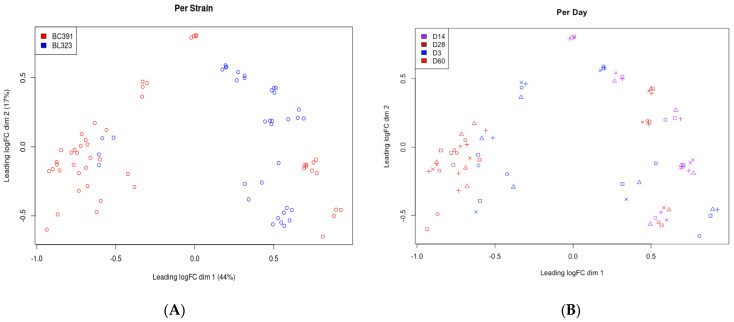
Multidimensional scaling analysis. See methods for details. (**A**) Graphical view of the variation of the data according to the strain type. Red circles correspond to gene expression data for strain BC-391. Blue circles correspond to gene expression data for strain BL-323. (**B**) Graphical view of the variation of the data according to the days post-infection. Each color represents the days post-infection as indicated by the conventions in the upper left box of the graph.

**Figure 5 ijms-23-05157-f005:**
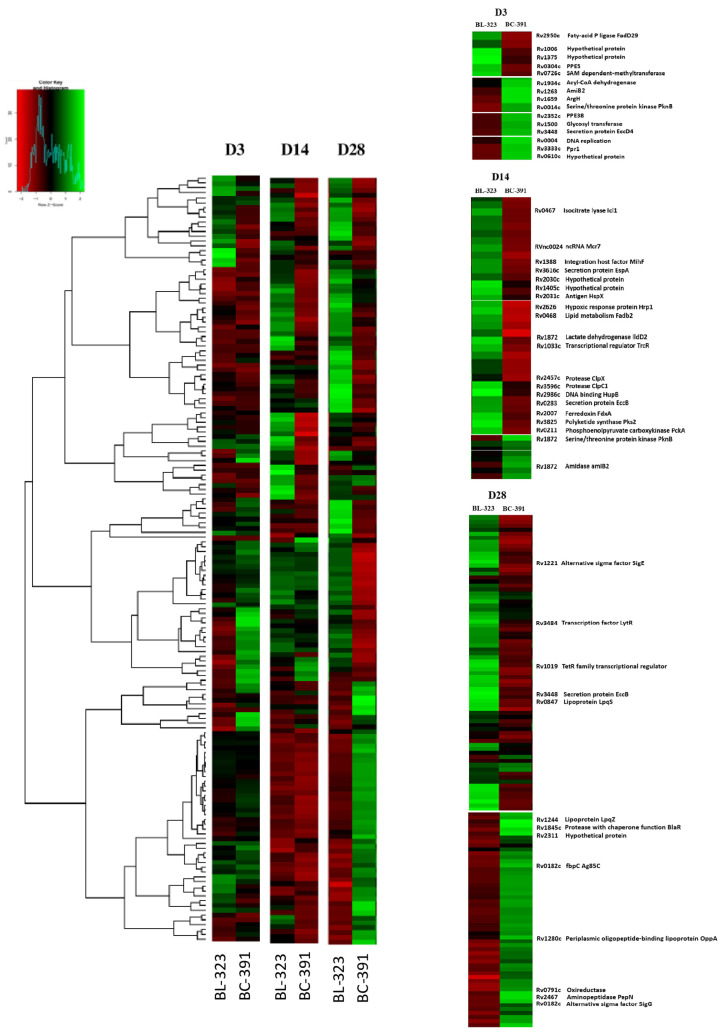
Comparative heatmap between strain BL-323 and BC-391 at each day of the analysis. The figure was edited to show the relevant data.

**Figure 6 ijms-23-05157-f006:**
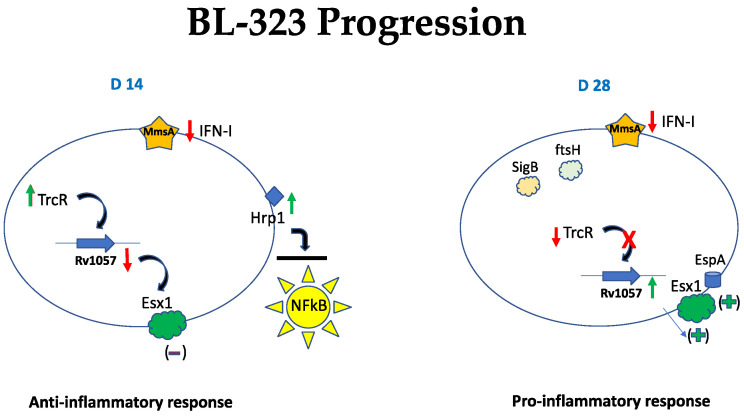
Schematical model describing key genes involved in progression of the infection in the murine model of the strain BL-323 at day 14 and 28 PI. Up and green arrows indicate over-expressed genes and down and red arrows indicate under-expressed genes.

**Figure 7 ijms-23-05157-f007:**
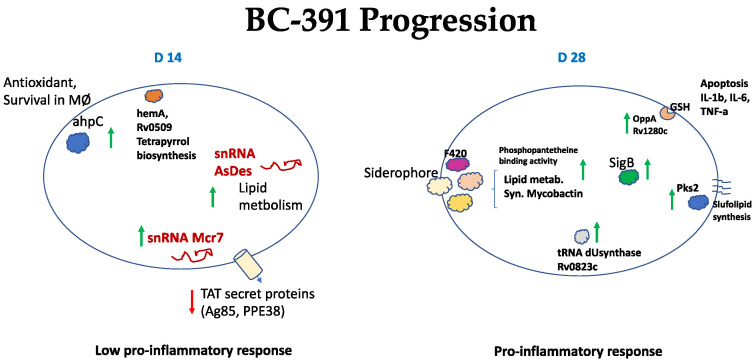
Schematical model describing key genes involved in progression of the infection in the murine model of the strain BC-391 at day 14 and 28 PI. Up and green arrows indicate over-expressed genes and down and red arrows indicate under-expressed genes.

**Figure 8 ijms-23-05157-f008:**
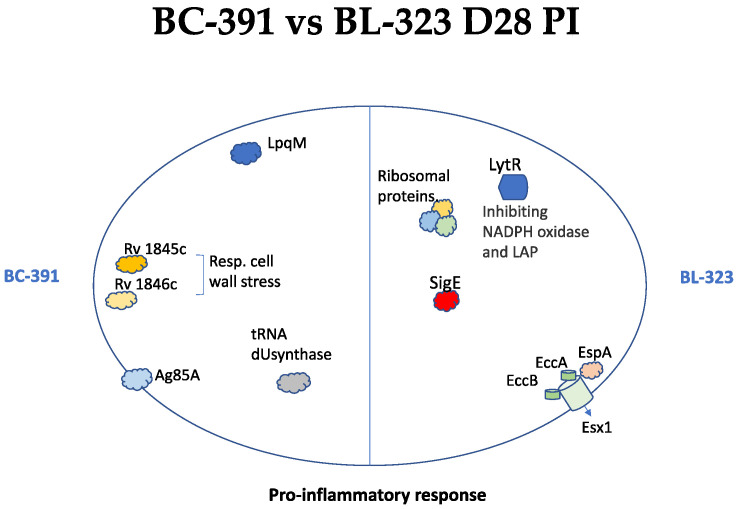
Schematic model describing key genes that lead to the pro-inflammatory response in both strains.

**Table 1 ijms-23-05157-t001:** Mapped reads to BL-323 and BC-391 genomes and genome cover.

Reads/Genes	BL-323 Day PI			BC-391 Day PI			
	3	14	28	3	14	28	60
**Total counts**	10,367,209.66 (100%)	9,861,857.58 (100%)	10,471,489.25 (100%)	12,374,677.58 (100%)	11,740,622.41 (100%)	12,301,584.5 (100%)	9,953,088.83 (100%)
**Not maped to mouse**	796,283.83 (7.68%)	875,917.66 (8.8%)	948,499.16 (9.06%)	757,210.5 (6.12%)	945,525.66 (8.05%)	809,753.83 (6.58%)	757,883.66 (7.61%)
**Maped to Mtb**	54,849 (0.53%)	65,719.91 (0.66%)	67,580.66 (0.64%)	51,140.83 (0.41%)	77,151.75 (0.65%)	60,099.25 (0.48%)	60,474.41 (0.60%)
**Maped to CDS Mtb ***	52,167.08 (95.1%)	58,889.41 (89.6%)	58,970.91 (87.26%)	49,105.41 (96.01%)	76,026.58 (98.54%)	56,595.16 (94.16%)	55,783.91 (92.24%)
**Maped to rRNA Mtb ***	2170.33 (3.95%)	5142.33 (7.8%)	6316 (9.34%)	1574.25 (3.07)	826 (1.07%)	3094.66 (5.15%)	4086.5 (6.75%)
**Maped to ncRNA***	28.58 (0.05%)	156.92 (0.24%)	277.58 (0.41%)	9.25 (0.02%)	82 (0.1%)	31.75 (0.05%)	170.41 (0.28%)
**Not associated ***	483 (0.88%)	1531.25 (2.33%)	2016.16 (2.98%)	430.91 (0.84%)	217.16 (0.28%)	377.66 (0.62%)	433.58 (0.71%)
**% of MTB genes mapped ****	1148 (30%)	1429 (37%)	1493 (39%)	1323 (33%)	705 (18%)	1357 (34%)	1623 (41%)

* Percentage of reads mapped regard total genes reported for each strain in NCBI database: BL-323: 3879 genes (100%); BC-391: 3906 genes (100%). ** Genes mapped against BL-323: 3879 genes (100%); BC-391: 3906 genes (100%).

## Data Availability

The complete sequence project was deposited in the GEO database, Accession Number: GSE 198877 https://www.ncbi.nlm.nih.gov/geo/query/acc.cgi?acc=GSE198877 accessed on 15 June 2021.

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
