# Peer review of "Close Related Drug-Resistance Beijing Isolates of Mycobacterium tuberculosis Reveal a Different Transcriptomic Signature in a Murine Disease Progression Model"

_ijms, 2022, doi:10.3390/ijms23095157_

Round 1

Reviewer 1 Report

Summary

In this article, the authors analysed the transcriptomic profile of two different Beijing genotypes in progressive TB murine model.

Major comments

I did not fully understand the differences between the study published by the same group (ref. 19) and the current study. The authors should clarify those differences, and also they should clarify earlier in the manuscript (and better) the fact that “the MTB transcriptomics reported here is part of a broader experiment”, and not only so late in the manuscript on page 10. Why the results of the broader experiment have not been published together? if this is not clarified, the added value of the present study is not clear. The experiment of in vivo transcriptomic analysis is the same as reported in ref 19, again please clarify the differences between the two studies (for example in the results section, section 2.1). Transcriptomic results are also reported similarly in ref.19 and in the current paper, i.e. transcriptomic profile of mice infected with strain BL-323, and transcriptomic profile of mice infected with strain  BC-391. Is the difference between the two studies consisting in the fact that the authors in the current study used the new RNAseq technology developed in their laboratory?

Minor comments

I was a bit confused regarding the naming of the Beijing family; sometimes it is referred to as the Beijing genotype, or family strains, or family, or isolates. Maybe the authors could be more consistent about that throughout the manuscript.

Page 2 Line 52: did you mean although scarce?

Page 2 Lines 60-61: I would suggest to re-phrase it in something like this: “in Cuba, Perú, and Colombia a high prevalence of multidrug-resistant (MDR) tuberculosis isolates has been reported

Page 2 Line 62: spell it out XDR here, the first time that you mention it, and not at line 64.

Page 2 Line 64: could you clarify the words “highly aggressive”? Did you mean more transmissible?

Page 2 Lines 68-71: I did not understand the genetic differences (if there are) between BL and BC; maybe the authors could clarify that? Maybe they could also explain the presence (and difference) between ancient and modern Beijing strains? I think slightly more background information about the Beijing family would be beneficial for the readers.

Page 2 Line 69: to enhance clarity and consistency, please replace “Beijing/W” with BC, if this is what you mean.

Page 2 Line 83: did you mean responses or RNA?

Page 2 Line 91: 30 amino acid or nucleotide deletion? PPE8 gene or protein? Please replace strains with the plural.

Page 10 Lines 299-300: please specify, e.g. clarify which strain is more virulent and how.

Page 16: please correct the numerical order of references

Reviewer 2 Report

This manuscript characterizes the BL-323 and BC-391 using transcriptomic assay. All experiments were well designed and the results obtained in this study was well interpreted, specifically the relevant genes involved in pro-inflammatory and anti-inflammatory responses. There are only minor comments on the format, such as Line 71 – delete parenthesis and page 16 – double check the reference numbers.

Round 2

Reviewer 1 Report

I thank the authors for addressing especially the major concern regarding the relation between the 2 studies from their group